# Regio- and stereoselective access to highly substituted vinylphosphine oxides via metal-free electrophilic phosphonoiodination of alkynes

Bingbing Dong[1], Fengqian Zhao[1], Wen-Xin Lv[2,3], Ying-Guo Liu [1],
Donghui Wei [1] ✉, Junliang Wu [1] ✉ & Yonggui Robin Chi [2,3] ✉

In general, the P-centered ring-opening of quaternary phosphirenium salts (QPrS) predominantly leads to hydrophosphorylated products, while the C-centered ring-opening is primarily confined to intramolecular nucleophilic reactions, resulting in the formation of phosphorus-containing cyclization products instead of difunctionalized products generated through inter-molecular nucleophilic processes. Here, through the promotion of ring-opening of three-member rings by iodine anions and the quenching of electronegative carbon atoms by iodine cations, we successfully synthesize $\beta$-functionalized vinylphosphine oxides by the P-addition of QPrS intermediates generated in situ. Multiple $\beta$-iodo-substituted vinylphosphine oxides can be obtained with exceptional regio- and stereo-selectivity by reacting secondary phosphine oxides with unactivated alkynes. In addition, a variety of $\beta$-func-tionalized vinylphosphine oxides converted from C-I bonds, especially the rapid construction of benzo[b]phospholes oxides, demonstrates the significance of this strategy.

Organophosphorus are a predominant class of organic compounds extensively present in fire retardants[1–4], pesticides[5–7], natural products, and biologically active molecules[8–11]. Among the various types of organophosphorus compounds, vinylphosphine oxides play a vital role as the fundamental synthetic component for the construction of these essential molecules[12,13]. The construction of Csp²-P is of great importance in the field of organophosphorus chemistry. Many elegant methods have been developed for the synthesis of these compounds[14–28], while the construction of $\beta$-functionalized vinylpho-sphine oxides remains highly restricted[29–33]. In the limited number of reports thus far, there is a frequent need for significant quantities of oxidants or metal salts. Considering the significance of $\beta$-

functionalized vinylphosphine oxides in the modification of organo-phosphorus skeleton, the pursuit of diverse approaches to their pre-paration using easily obtainable substrates remains highly appealing.

On the other hand, the value of three-membered rings possessing a heteroatom in organic transformations has been demonstrated by the rapid and atomically economical synthesis of $\beta$-functionalized alcohols and amines via ring opening processes[34,35]. Quaternary phosphirenium salts (QPrS), which could be easily generated from alkynes and secondary phosphine oxides in the presence of Tf₂O, are also considered to be powerful intermediates for the construction of organophosphorus compounds[36–42]. However, in contrast to epoxides[43–47] and aziridines[48–51], the phosphorus atom possesses

[1]Green Catalysis Center, and College of Chemistry, Zhengzhou University, Zhengzhou 450001, PR China. [2]National Key Laboratory of Green Pesticide, Key Laboratory of Green Pesticide and Agricultural Bioengineering, Ministry of Education, Guizhou University, Guiyang 550025, PR China. [3]School of Chemistry, Chemical Engineering, and Biotechnology, Nanyang Technological University, Singapore 637371, Singapore. ✉e-mail: donghuiwei@zzu.edu.cn; wujl@zzu.edu.cn; robinchi@ntu.edu.sg

inherent electrophilicity, resulting from the polarity inversion of the C-P bond[52], which leads to C-addition occurring exclusively when a weakly nucleophilic reagent attacks the ring[53–55] (Fig. 1a). When employing potent nucleophilic reagents, P-addition will be prioritized to afford hydrophosphorylated products instead of β-functionalized vinylphosphine oxides. For example, Wild's group reported that phosphirenium triflates can undergo ring-opening to obtain cis-hydrophosphorylated products in the presence of MeOH or $H_2O$[52]. To the best of our knowledge, there are only a few examples of β-substituted phosphine compounds obtained via C-addition for example using aniline[56] or intramolecular aromatic as nucleophilic reagents[53], while the synthesis of these compounds through a P-addition ring-opening process remains unreported.

Although this completive ring-opening process leads to incompatibility of strongly nucleophilic reagents in the synthesis of β-functionalized vinylphosphine oxides, it also presents an additional opportunity for the synthesis of these compounds. When the P-centered nucleophilic addition of quaternary phosphirenium salts is occurring due to nucleophilic reagents attacks on the phosphine, the electronegative carbon atoms could therefore be trapped by electrophilic reagents (Fig. 1b). Therefore, by coordinating the presence of these two reagents in the system, it becomes possible to synthesize β-substituted vinylphosphine oxides through P-addition process. Iodine monomers are known to readily polarize in solvents, producing iodine anions and cations, which may be utilized in our designs as

nucleophilic and electrophilic reagents respectively. Therefore, the regio- and stereoselective phosphonoiodination of unactivated alkynes have been described herein (Fig. 1c). The attack of iodine nucleophilic reagents on the phosphorus atom leads to ring-opening of the three-membered ring while quenching of the carbon anion by iodine electrophilic reagents is essential for the production of β-functionalized vinylphosphine oxides. In addition, a variety of β-functionalized vinylphosphine oxides could be easily obtained through the conversion of C–I bonds. Furthermore, the skeleton of benzo[*b*]phosphole oxides could also be constructed through a radical cyclization process involving this particular class of compounds.

## Results and discussion
### Reaction development
According to our design, diphenylphosphine oxide **1a** and unactivated alkyne **2a** were selected as model substrates to produce quaternary phosphirenium salts in situ in the presence of $Tf_2O$. When $I_2$ was introduced at the start of the reaction, and the system was allowed to react in $CHCl_3$ (2.0 mL) at 60 °C for 21 h, no desired phosphonoiodination product was obtained (Table 1, entry 1). Considering that the presence of iodine might disrupt the production of phosphirenium intermediate from electrophilic phosphination reagent generated in situ and alkyne **2a**, we conducted the experiment without iodine for 3 h. Subsequently, we introduced iodine and allowed the reaction to proceed for an additional 18 h. To our delight, **3a** was isolated in 50%

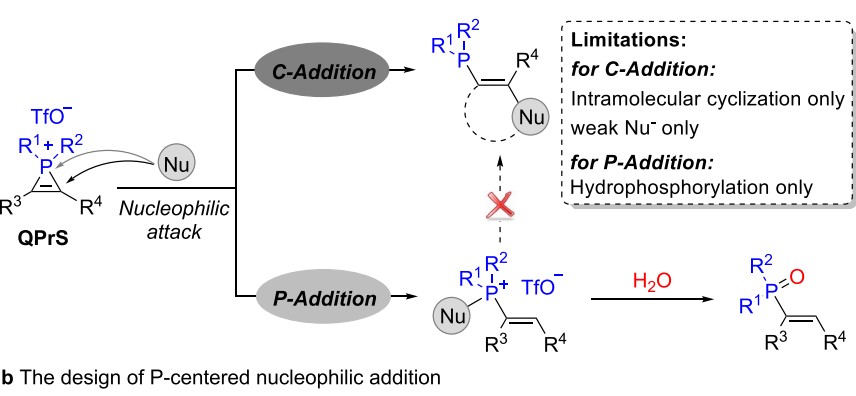

**a** Regioselectivity of nucleophilic ring-opening in QPrS

**b** The design of P-centered nucleophilic addition

**c** Regio- and stereoselective phosphonoiodination of unactivated alkynes *(this work)*

✔ P-centered nucleophilic addition

✔ Excellent regio- and stereo-selectivity

✔ β-Iodo-substituted vinylphosphine oxides

✔ Synthetic values for various P-containing compounds

**Fig. 1 | Reaction design for the synthesis of β-functionalized vinylphosphine oxides. a** Regioselectivity of nucleophilic ring-opening in QPrS. **b** The design of P-centered nucleophilic addition. **c** Regio- and stereoselective phosphonoiodination of unactivated alkynes (this work).

**Table 1 | Optimization of reaction conditions[a]**

| Entry | Activating species | Base | I source | Yield of 3a (%)[b] |
|---|---|---|---|---|
| 1[c] | Tf$_2$O | / | I$_2$ | trace |
| 2 | Tf$_2$O | / | I$_2$ | 50 |
| 3 | Tf$_2$O | BDMEP | I$_2$ | 69 |
| 4 | Tf$_2$O | BDMEP | I$_2$ | 32 |
| 5 | Tf$_2$O | DBU | I$_2$ | trace |
| 6 | TMSOTf | BDMEP | I$_2$ | n.d. |
| 7 | BF$_3$·Et$_2$O | BDMEP | I$_2$ | n.d. |
| 8 | (CF$_3$CO)$_2$O | BDMEP | I$_2$ | n.d. |
| 9 | Tf$_2$O | BDMEP | NIS | 33 |
| 10 | Tf$_2$O | BDMEP | ICl | 27 |
| 11 | Tf$_2$O | BDMEP | TBAI/NaI | n.d. |
| 12[d] | Tf$_2$O | BDMEP | I$_2$ | 66 |
| 13[e] | Tf$_2$O | BDMEP | I$_2$ | 35 |
| 14[f] | Tf$_2$O | BDMEP | I$_2$ | 83 |

"n.d." stands for "not detected".

*Tf$_2$O* trifluoromethanesulfonic anhydride, *TMSOTf* trimethylsilyl trifluoromethanesulfonate, *BDMEP* 2,6-Di-*tert*-butylpyridine, *DBU* 1,8-Diazabicyclo[5.4.0]undec-7-ene, *NIS N*-Iodosuccinimide, *TBAI* Tetrabutylammonium iodide.

[a]Reaction conditions: **1a** (0.20 mmol, 1.0 equiv), **2a** (0.40 mmol, 2.0 equiv), activating species (0.20 mmol, 1.0 equiv), base (0.40 mmol, 2.0 equiv), CHCl$_3$ (2.0 mL), 60 °C, 3 h, then I source (0.40 mmol, 2.0 equiv) was added for 18 h. [b]Isolated yield.
[c]I source was added at first and stirred at 60 °C for 21 h.
[d]**2a** (0.30 mmol, 1.5 equiv) was used.
[e]**2a** (0.20 mmol, 1.0 equiv) was used.
[f]CHCl$_3$ (1.0 mL) and Tf$_2$O (0.21 mmol, 1.05 equiv) were used.

yield (Table 1, entry 2), and the structure was confirmed by X-ray crystallographic analysis (CCDC no. 2240040). When 2.0 equiv of 2,6-di-*tert*-butylpyridine was added as a base, the yield of **3a** could be increased to 69% (Table 1, entry 3). Other bases, such as 2,4,6-trimethylpyridine and Na$_2$HPO$_4$, gave a significant decrease in yield, and no target product was detected when Et$_3$N, DBU, DABCO, K$_2$HPO$_4$, Na$_2$CO$_3$ were employed as the base (see Supplementary Information for more details). Additionally, no desired product was obtained when Tf$_2$O was replaced by other activating species such as TMSOTf, BF$_3$·Et$_2$O, and (CF$_3$CO)$_2$O (Table 1, entries 6–8). Although NIS and ICl were considerably less efficient, affording **3a** in only 33% and 27% yields, respectively, the screening of different I source indicated that an I$^+$ source was crucial for successful phosphonoiodination of alkynes, which may also prove that the process was not carried out through C-addition (Table 1, entries 9–11). No product was detected when the reaction was performed at room temperature, and the yield was not improved by increasing the temperature (see Supplementary Information for more details). There was no significant effect on the reaction by slightly reducing the amount of alkyne (Table 1, entry 12). Finally, the yield of **3a** was significantly increased to 83% yield by slightly increasing the concentration of the reaction and the amount of Tf$_2$O (Table 1, entry 14).

## Substrate scope

With the optimized conditions in hand, we first begin testing our substrate scope with secondary phosphine oxides. As shown in Fig. 2, diarylphosphine oxides bearing electron-withdrawing and electron-donating groups on the benzene ring were suitable for this reaction, delivering the corresponding products in moderate to good yields (**3a**–**3n**). The steric effect had a certain impact on the reaction, leading

to decreased yields when the benzene rings had substituents at the *meta*-position (**3f, 3h**–**3i**). Substituents at the *ortho*-position of the benzene rings resulted in the major products being trivalent phosphorus (**3m'** and **3n'**), with a certain amount of pentavalent phosphorus. This result may be attributed to steric hindrance that leads to the C-centered ring-opening. And we didn't observe trivalent phosphorus products when using other SPOs as substrates. To determine the *Z/E* configuration of trivalent phosphorus products, we chose **3m'** as an example. It was oxidized to pentavalent phosphorus by H$_2$O$_2$, and the configuration was subsequently confirmed through X-ray crystallographic analysis (CCDC no. 2300801). In addition to diarylphosphine oxide, cyclopentyl(phenyl)phosphine oxide was also suitable substrate, providing the corresponding product **3l** in 45% yield.

Subsequently, we examined the scope of alkynes in the phosphonoiodination reactions. As shown in Fig. 3, a variety of functional groups, such as methyl (**4a**–**4b**), methoxy (**4c**–**4d**), methylthio (**4e**), phenyl (**4f**), trimethylsilyl (**4g**), fluorine (**4h**), chlorine (**4i**), trifluoromethyl (**4j**–**4k**) and trifluoromethoxy (**4l**) were all tolerated in the reaction. When an electron-withdrawing group was attached to the benzene ring, the yield decreased rapidly as the electron-withdrawing effect increases (**4j**–**4k**), which may be because the electron-withdrawing group is unfavorable for the formation of the three-membered cyclic phosphorus cation intermediate, and only phosphoric anhydride and products were observed in the analysis of [31]P NMR spectra of the crude mixture (See Supplementary Fig. 2, Supplementary Information for more details). Meanwhile, disubstituted phenyl (**4m**), naphthyl (**4n**), and other heterocycles such as benzothienyl (**4o**), benzofuranyl (**4p**), and thiophenyl (**4q**) can be successfully transformed in the system. Furthermore, alkynes with various functional group substitutions on the carbon chain were also well

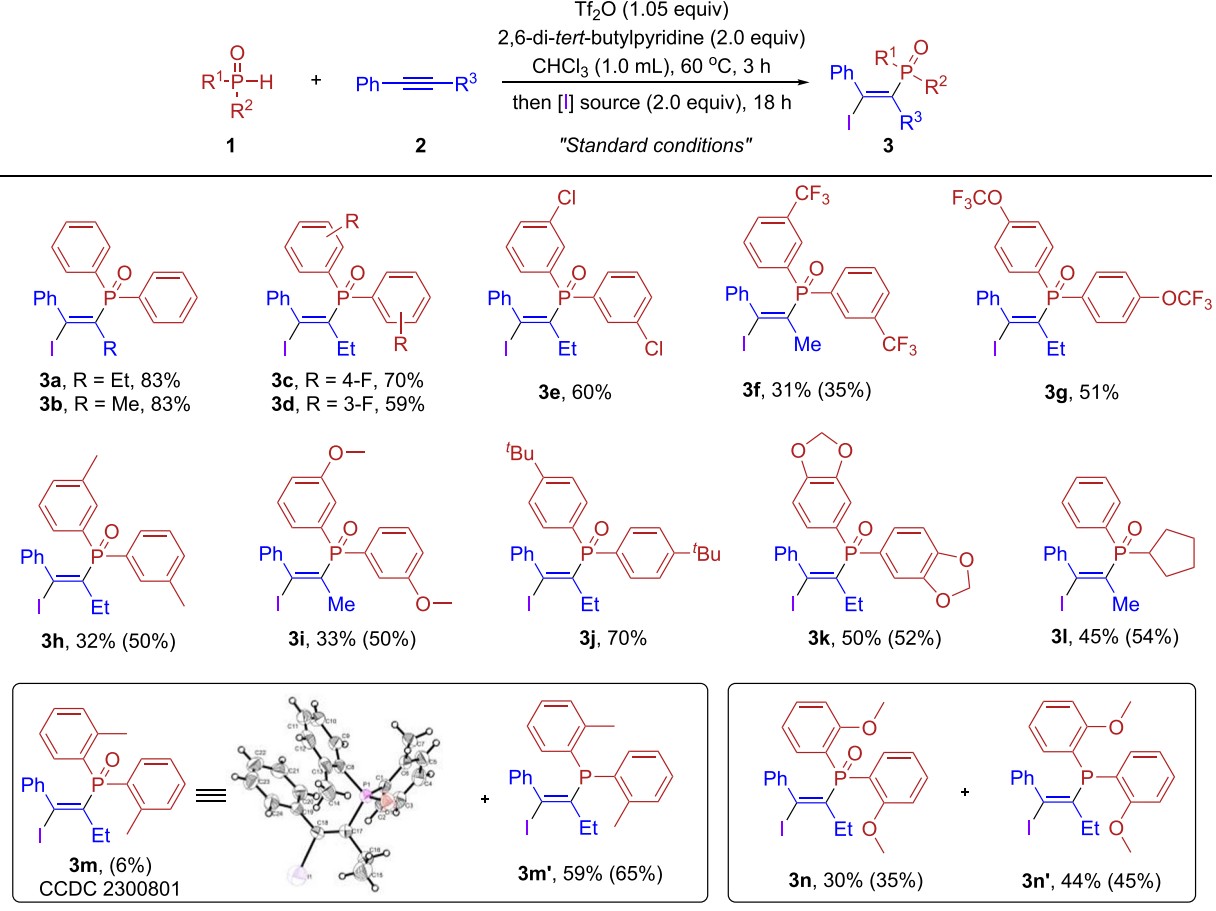

**Fig. 2 | Scope of secondary phosphine oxides in phosphonoiodination.** Reaction conditions: **1** (0.20 mmol, 1.0 equiv), **2a** or **2b** (0.40 mmol, 2.0 equiv), Tf$_2$O (0.21 mmol, 1.05 equiv), 2,6-di-*tert*-butylpyridine (0.40 mmol, 2.0 equiv), CHCl$_3$ (1.0 mL), 60 °C, 3 h, then I$_2$ (0.40 mmol, 2.0 equiv) was added for 18 h. Isolated yield and the number in parentheses is $^1$H NMR yield based on dimethyl terephthalate.

compatible in this system, and the presence of ketones (**4r**), three-membered rings (**4s**), double bonds (**4t**) and protected amino groups (**4u**) offer further possibilities for subsequent modifications. To our delight, aliphatic internal alkynes, such as hex-3-yne could also be successfully employed to afford the desired phosphonoiodination products **4v**, albeit in only 15% yield.

## Mechanistic study

To gain more insight into the mechanism of this reaction, several mechanistic experiments were performed. As shown in Fig. 4, the isotope labeling experiments were performed firstly. When the reaction was carried out under standard conditions using diphenylphosphine with an oxygen isotope label and quenched with water, there was no isotope label in the product. In contrast, when the model reaction was quenched with oxygen-labeled water, product with oxygen isotope labeling was obtained in 60% yield (Fig. 4a). These results indicate that the oxygen present in the product originates solely from water. Subsequently, when the substrate **5** was used in the reaction and iodine was added at the beginning, both the phosphonoiodination product **6** and the cyclization product **7** were obtained simultaneously (Fig. 4b)[53]. This suggests that the phosphonoiodination reaction may also undergo a three-membered cyclic phosphorus cation intermediate and that these products result from a competitive reaction between the P-addition and C-addition process of this intermediate. To further confirm the existence of this intermediate in our reaction, we conducted an in situ $^{31}$P NMR experiment before adding I$_2$ and observed a distinct peak of −104.4 ppm which is assigned to phosphirenium species **A** (Fig. 4c).

Additionally, density functional theory (DFT) calculations were employed to elucidate the possible mechanism and chemoselectivity of the ring-opening process. In Fig. 4d, DFT calculations based on phosphirenium iodide indicated that both kinetics and thermodynamics support the generation of the product. Further calculations on phosphirenium triflate were performed, revealing that the energy barrier for the transition states is significantly higher than that of the ring-opening of phosphirenium iodide (See Supplementary Fig. 4, Supplementary Information for more details). Therefore, we thought that the presence of the iodide increases the driving force for the ring-opening. Moreover, we considered the direct cleavage of the I–I bond, locating a concerted transition state **A-TS3** (See Supplementary Fig. 5, Supplementary Information for more details), in which the cleavage of C–P bond is coupled with the formation of C–I and P–I bonds. However, the energy barrier is prohibitively high, so this pathway involving the direct cleavage of I–I bond can be excluded safely.

## Synthetic applications

To further demonstrate the synthetic value of this procedure, transformations of product **3a** and **3b** were carried out. As shown in Fig. 5, various β-functionalized vinylphosphine oxides can be obtained in excellent yields by conversion of the C-I bond. For instance, **3a** was easily transformed to tetrasubstituted vinylphosphine oxides containing azido (**8**), cyano (**9**) and pyrrolo groups (**10**) at the β-position. Hydrodehalogenation of **3a** in the Zn/acetic acid system afforded the trisubstituted vinylphosphine oxides **11** (*Z/E* = 2:1) in excellent yield. The corresponding vinylphosphine oxides **12** and **13** were also produced with retention of the olefin stereochemistry through

**Fig. 3 | Scope of alkynes in phosphonoiodination.** Reaction conditions: **1a** (0.20 mmol, 1.0 equiv), **2** (0.40 mmol, 2.0 equiv), Tf$_2$O (0.21 mmol, 1.05 equiv), 2,6-di-*tert*-butylpyridine (0.40 mmol, 2.0 equiv), CHCl$_3$ (1.0 mL), 60 °C, 3 h, then I$_2$ (0.40 mmol, 2.0 equiv) was added for 18 h. Isolated yield.

Suzuki–Miyaura and Stille couplings, respectively. Additionally, Heck coupling of **3a** afforded the conjugate vinylphosphine oxide **14** with a mixture of isomers (*Z, E/Z, Z* = 5:4).

In addition to transition metal-catalyzed coupling reactions, C-I bonds can also be transformed by other types of reactions. According to the literature, homolytic cleavage of C-I bonds in the presence of strong electron-absorbing groups can generate corresponding iodine radicals and carbon radicals[23,57–61]. Here, we achieved a homogeneous cleavage of the C–I bond in the presence of 8-hydroxyquinoline and Cs$_2$CO$_3$, and the resulting carbon radical can undergo radical cyclization to give the benzo[*b*]phosphole oxides. As shown in Fig. 6, benzo[*b*]phosphole oxides with a variety of alkyl, aryl and heteroaryl groups on the backbone can be obtained in excellent yields (**15a–15i**). When there was methoxyphenyl substitution at the *α*-position of the carbon radical, the yield of the target product decreased significantly (**15j**), meanwhile, the by-product ketone **16** was obtained in 31% yield. Besides, we are also exploring other transformations for this alkenyl radical containing a vinylphosphine oxide group.

In conclusion, we have described a strategy for the synthesis of *β*-iodo-substituted vinylphosphine oxides from readily available secondary phosphine oxides, alkynes, and iodine, which was achieved by activation of the alkyne through Ar$_2$P-OTf generated in situ to form three-membered phosphorus cation intermediate, followed by ring-opening with I$_2$. The successful implementation of this strategy for synthesizing *β*-functionalized vinylphosphine oxides through P-addition heavily relied on the precise coordination between electrophilic and nucleophilic reactivity of iodine. The reaction was metal-free and did not necessitate the presence of additional oxidants.

Various *β*-functionalized vinylphosphine oxides can be obtained by the conversion of the C–I bonds. Finally, the construction of the benzo[*b*] phosphole oxide skeleton can be achieved by homologous cleavage of the C–I bonds and radical cyclization.

## Methods
### General procedure for the preparation of 3 and 4
A 10 mL oven-dried sealed tube equipped with a magnetic stir bar was charged with secondary phosphine oxides **1** (0.2 mmol, 1.0 equiv), alkynes **2** (0.4 mmol, 2.0 equiv, if solid). The tube was evacuated and backfilled with argon (three times) and then CHCl$_3$ (1.0 mL) was added sequentially via a syringe, followed by alkynes **2** (0.4 mmol, 2.0 equiv, if oil), 2,6-di-*tert*-butylpyridine (0.4 mmol, 2.0 equiv) and Tf$_2$O (0.21 mmol, 1.05 equiv) were added by a syringe. The resulting mixture was stirred for 3 h at 60 °C, and I$_2$ (0.4 mmol, 2.0 equiv) was added for 18 h at 60 °C. After cooled to ambient temperature, sat. NaHCO$_3$ aq (5.0 mL) was added and the resulting mixture was extracted with DCM (3 × 10 mL). The organic layer was washed with sat. Na$_2$S$_2$O$_3$ aq and brine, followed by dried over MgSO$_4$, and volatiles were removed under reduced pressure. The residue was purified by flash column chromatography on silica gel to give the desired products **3** and **4**.

### General procedure for the preparation of 15
A 10 mL oven-dried sealed tube equipped with a magnetic stir bar was charged with **3** or **4** (0.10 mmol, 1.0 equiv), 8-hydroxyquinoline (0.02 mmol, 0.2 equiv) and Cs$_2$CO$_3$ (0.2 mmol, 2.0 equiv). The tube was evacuated and backfilled with argon (three times) and then DMF (2.0 mL) was added sequentially via a syringe. The resulting mixture

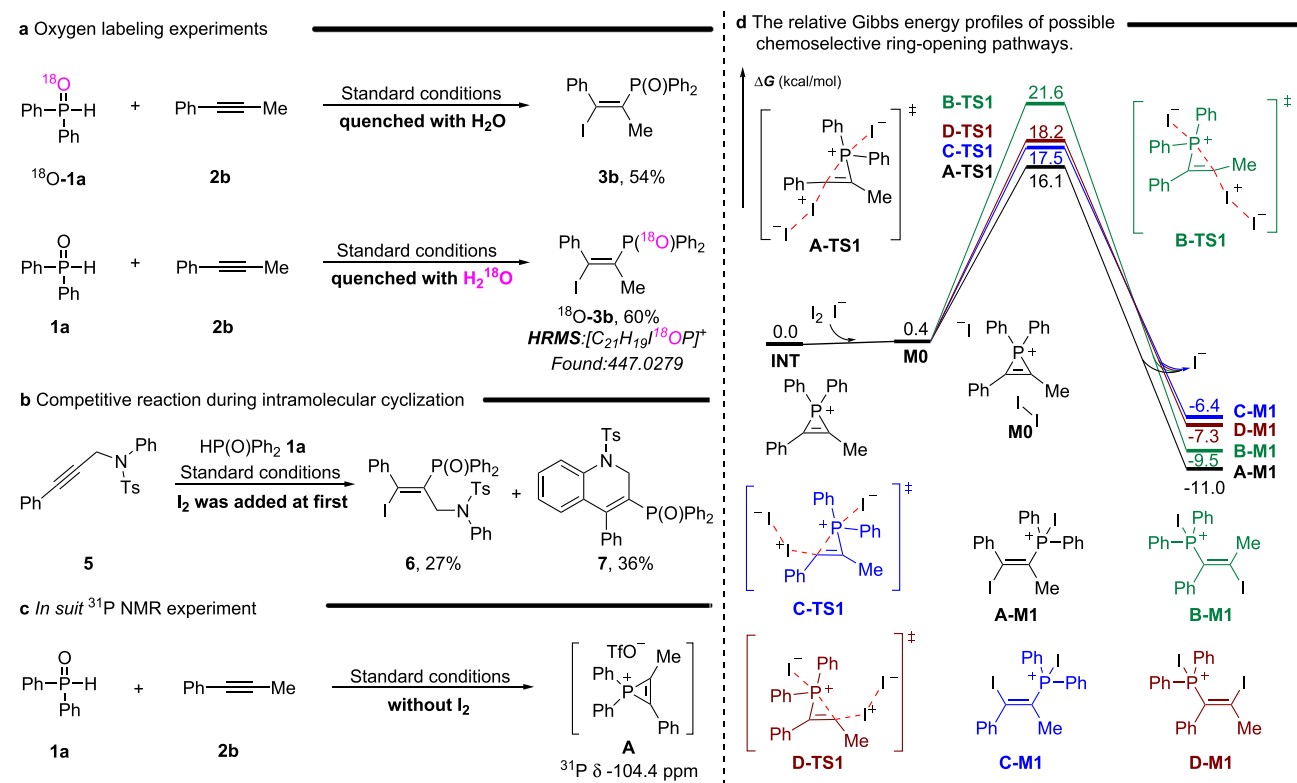

**a** Oxygen labeling experiments

**b** Competitive reaction during intramolecular cyclization

**c** In suit ³¹P NMR experiment

**d** The relative Gibbs energy profiles of possible chemoselective ring-opening pathways.

**Fig. 4 | Mechanistic experiments and DFT calculations. a** Oxygen labeling experiments. **b** Competitive reaction during intramolecular cyclization. **c** In situ ³¹P NMR experiment. **d** The relative Gibbs energy profiles of possible chemoselective ring-opening pathways.

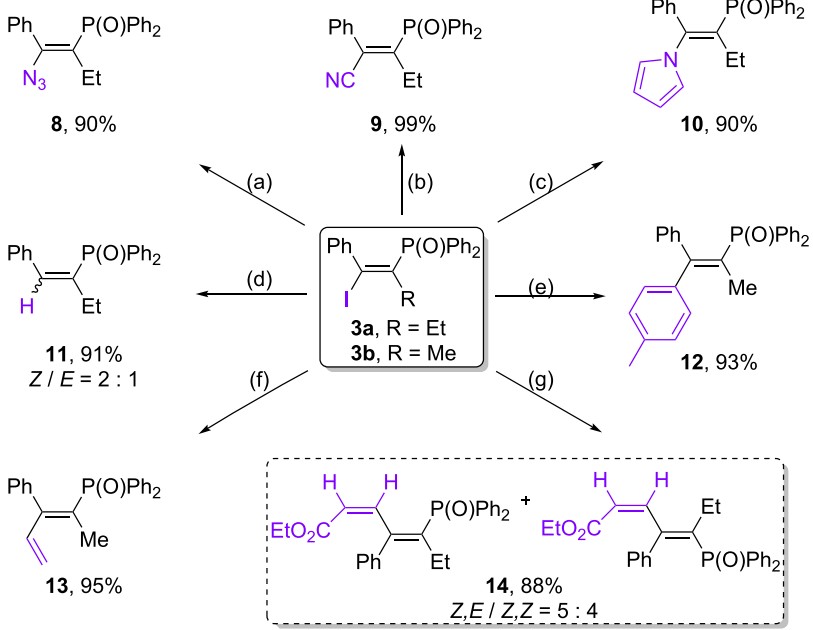

**Fig. 5 | Derivatization of the C-I bond of β-iodo vinylphosphine oxides 3.**
Reaction conditions: (**a**) $NaN_3$, DMAC; (**b**) CuCN, DMF; (**c**) pyrrole, CuI, DMEDA, $K_3PO_4$, toluene; (**d**) Zn, $H_2O$/AcOH; (**e**) 4-methylphenylboronic acid, $Pd_2(dba)_3$, X-Phos, $Cs_2CO_3$, DMF/$H_2O$; (**f**) tributyl(ethenyl)stannane, $Pd(PPh_3)_2Cl_2$, DMF; (**g**) methyl acrylate, $Pd(OAc)_2$, TBAB, $NaHCO_3$, DMF.

was stirred for 12 h at 110 °C. After cooled to ambient temperature, $H_2O$ was added and the resulting mixture was extracted with EA (3 ×10 mL). The organic layer was washed with brine and dried over $MgSO_4$, and volatiles were removed under reduced pressure. The residue was purified by flash column chromatography on silica gel to give the desired products **15**.

## Data availability
The X-ray crystallographic coordinates for structures reported in this study have been deposited at the Cambridge Crystallographic Data Center (CCDC), under deposition numbers CCDC 2240040 (**3a**), and CCDC 2300801 (**3m**). These data can be obtained free of charge from The Cambridge Crystallographic Data Center via www.ccdc.cam.ac.uk/

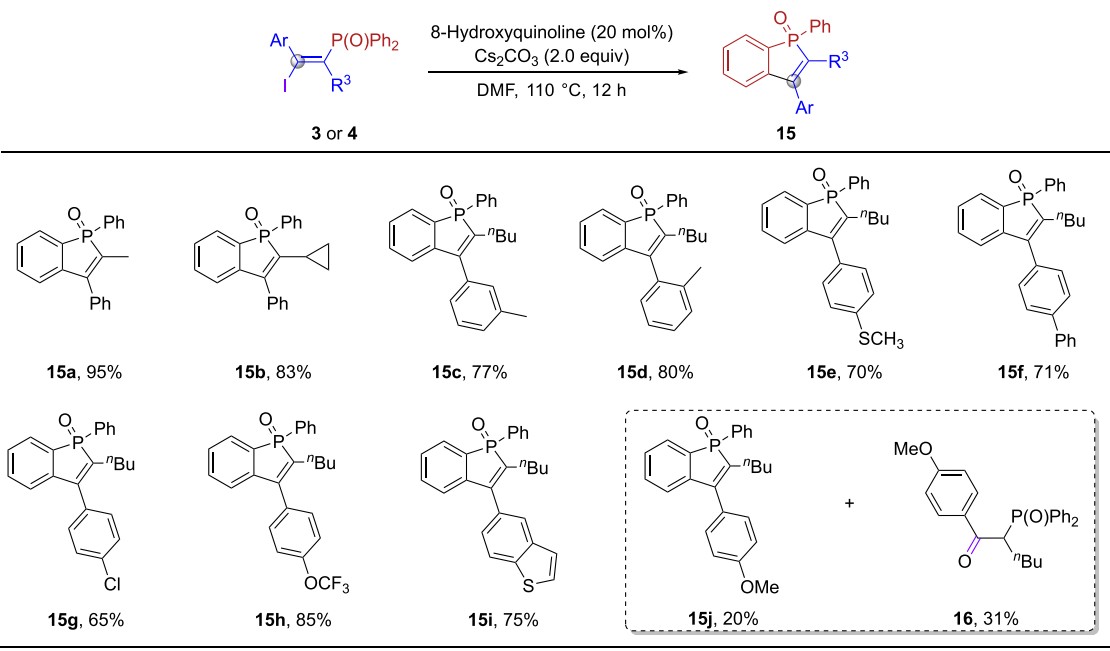

**Fig. 6 | Synthesis of benzo[*b*]phosphole oxides by radical cyclization of 3.** Reaction conditions: **3** or **4** (0.10 mmol, 1.0 equiv), 8-hydroxyquinoline (0.02 mmol, 20 mol %), Cs$_2$CO$_3$ (0.2 mmol, 2.0 equiv), DMF (2.0 mL), 110 °C, 12 h. Isolated yield.

data_request/cif. The full experimental details for the preparation of all new compounds, and their spectroscopic and chromatographic data generated in this study are provided in the Supplementary Information. All data are available from the corresponding author upon request. Source data are present. Source data are provided with this paper.

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

## Acknowledgements

This work was supported by the Top Youth Talent Fund of Zhengzhou University (J.W.); National Natural Science Foundation of China (No. 22101266, Y.-G.L.), and Postdoctoral Research Grants of Henan Province ([2023] 22120051, F.Z.).

## Author contributions

B.D. designed and conducted the experimental protocols and analyzed the experimental results. F.Z. and W.-X.L. wrote the initial draft and provided constructive advice. Y.-G.L. and D.W. conducted the DFT

studies. J.W. and Y.R.C. supervised the research and revised the manuscript with comments from all authors.

## Competing interests

The authors declare no competing interests.
