## [Peer Review File · Nature Communications]

Regio- and Stereoselective Access to Highly Substituted Vinylphosphine Oxides via Metal-Free Electrophilic Phosphonoiodination of AlkynesReviewers' Comments:

Reviewer #1:

Remarks to the Author:

The manuscript by Chi, et al describes the synthesis of β -iodo vinylphosphine oxides through activation of secondary phosphine oxides with Tf₂O, followed by addition on disubstituted alkynes and opening of the resulting phosphirenium ions intermediates with iodine. The article is generally well written and the supporting information is well done with all new compounds suitably characterized and described. The substrate scope has been studied thoroughly and the methodology is found to be widely applicable. Additional examples from non-aryl phosphine oxides (only one example shown) might nevertheless have been beneficial.

Without any doubt, the manuscript is a consistent and interesting paper, however, I consider that major concerns should be addressed prior to publication.

The title is not appropriate : It promotes the use of quaternary phosphiranium salts while the chemistry presented in the manuscript is not based on, nor involves, the formation of phosphiranium ions which are strictly speaking unsaturated three-membered cyclic phosphorus cations. Moreover, this suggests that the reaction is carried out using isolated species, which is not the case.

In addition, the confusion between the two phosphirenium and phosphiranium species is also apparent at several occasions in the manuscript and must be necessarily corrected.

Right from the abstract, the authors state that the opening is limited to intramolecular reactions leading only to the formation of cyclic phosphorus compounds. However, there are prior examples, mentioned later in the article (ref 50 and 54) but not clearly identified as such, of intermolecular reactions involving the opening of phosphiranium ions and phosphirenium ions by external nucleophiles. Ref 50 is cited to illustrate electrophilicity of phosphorus in three-membered phosphorus cationic species. And indeed, in this work, ring opening was achieved through addition of water and primary alcohols on phosphorus to produce the corresponding ring-opened products. This therefore clearly constitutes a precedent for the addition of a nucleophile to P, and the authors must therefore refer to it in an appropriate and unambiguous way. Concerning ref 54, a C-centered ring opening of isolated phosphiranium salts by anilines, leading to β -functionalized linear phosphines, is described. In connection with this work, the authors point out that the quoted reaction leads to the generation of phosphine oxides (line 54, page 3), whereas the work described by the French group produces unoxidized phosphines.

Two additional articles reporting electrophilic phosphination of alkynes promoted by Tf₂O should be added in ref 40 :

(1) Hirano, K. and Miura, M. Development of New C-N and C-P Bond Formations with Alkenes and Alkynes Based on Electrophilic Amination and Phosphination. *J. Synth. Org. Chem. Jpn.* 2018, 1206-1214.

(2) Kazutoshi Nishimura, K. ; Xu, S. ; Nishii, Y. and Hirano, K. One-Step Synthesis of Benzophosphole Derivatives from Arylalkynes by Phosphirenium-Dication-Mediated Sequential C-P/C-C Bond Forming Reaction. *Org. Lett.* 2023, 25, 1503–1508.

In this regard, ref 51 should not be disconnected from these latter major contributions since the chemistry developed therein is strongly correlated. This same reference should also be recalled when commenting on the results obtained from a competition reaction test using a propargylamine substrate (line 144, page 6).

Concerning mechanistic consideration, it would have been useful if the authors had mentioned whether they had actually observed the formation of phosphirenium intermediates, in particular by monitoring the reaction using ³¹P NMR, since phosphirenium ions exhibit a characteristic shift around -110 ppm. If not, control experiments should be carried out.

I also deplore the absence of discussion on the regioselectivity of the reaction observed with disymmetrical alkynes, especially since DFT calculations that constitute elements of discussion are succinctly given in the experimental section. At the very least, comments on this point should be included in the article and calculations should be further commented.

Besides, the literature is not always cited in an appropriate manner. I recommend that authors

rigorously check the references of the all the articles cited. See below for representative examples of errors found :

-Ref 29 (line 249, page 9): No journal mentioned

-Ref 40 (line 274, page 10): The name of the Journal is wrong . The cited article is actually published in Chemistry a european Journal (Chem. Eur. J. 2018, 24, 13089 – 13092).

-In some references, the authors are not fully cited whereas they are in other cases.

A few misspelling or typing errors were also detected :

-Difunctionalization (« c » missing) in the title (line 2, page 1)

-« is occur » should be replaced by « is occurring » (line 60, page 3)

-« condition » should be replaced by « conditions » (line 97, page 4)

-« cynao » should be replaced by « cyano » and « pyrro » by « pyrrolo groups » (line 151, page 6)

-« iodines » should be replaced by « iodine » (line 181, page 8)

-« reagents » should be replaced by « reactivity » (line 186, page 8)

-« phosphorhole » should be replaced by « phosphole » (line 188, page 8)

In summary, I could not recommend the current manuscript for publication as it is. However, since the work presented in the article is interesting and should be of interest for synthetic chemists, I believe that it could be reconsidered for publication in Nature Communications provided that major revisions are made to the article.

Reviewer #2:

Remarks to the Author:

Chi and coworkers present a method for the preparation of E-iodo-vinylphosphine oxides using elemental iodine as the source of I.

The manuscript is well laid out and easy to read.

The reaction operates over a range of phosphine oxide starting materials, with several sterically and electronically interesting and diverse phosphine oxides used. In this regard the authors have gone beyond a routine substrate scope. This is also demonstrated in the range of alkynes probed in the reaction.

The authors report isolated yields for their products, which is great, but this actually negatively impacts the impression of the research, which half of Figure 1 being yields that are 50% or less... 4 of the yields are in the low 30% range. It would be of value to readers for the authors to report a spectroscopic yield of these products and to comment on what else forms e.g. is the remaining 50-70% across these substrates (3f, h, i, j, k, m, n) unreacted starting material?

For the mechanistic studies- this reviewer feels that it is generally accepted that secondary phosphine oxide + Tf₂O + alkyne generates, as termed by the authors, QPrS and other research groups have isolated these 3-membered rings. So although their studies confirm this (Fig4a), this does not bring any particularly new insight.

Fig 4b- is there evidence for in situ QPrS with a chloride counterion? This study, in its current form, does not rule out another addition mechanism taking place for the chlorophosphine.

Fig4c- is this not a repeat of Miura's work (which we know proceed through the 3 membered QPrS) and therefore does not add anything new? Maybe I have misunderstood what the authors are trying to demonstrate here?

Page 6, line 156/157- some references to back up this statement are needed.

Figure 6 (and associated discussion)- one could argue that this method is less economical than the methods from Chen et al. and from Liu et al. because their routes simply involve the reaction of

secondary phosphine oxide with alkyne, rather than the need of an iodide to be introduced and then lost- a more critical evaluation of their own work is needed here for context.

All in all, there's some interesting synthesis presented here, but there are details lacking and poor representation of the state-of-the-art in this field. Both aspects need worked on.

REVIEWER COMMENTS

Reviewer #1 (Remarks to the Author):

The manuscript by Chi, et al describes the synthesis of β -iodo vinylphosphine oxides through activation of secondary phosphine oxides with Tf₂O, followed by addition on disubstituted alkynes and opening of the resulting phosphirenium ions intermediates with iodine. The article is generally well written and the supporting information is well done with all new compounds suitably characterized and described. The substrate scope has been studied thoroughly and the methodology is found to be widely applicable. Additional examples from non-aryl phosphine oxides (only one example shown) might nevertheless have been beneficial.

Without any doubt, the manuscript is a consistent and interesting paper, however, I consider that major concerns should be addressed prior to publication.

The title is not appropriate: It promotes the use of quaternary phosphiranium salts while the chemistry presented in the manuscript is not based on, nor involves, the formation of phosphiranium ions which are strictly speaking unsaturated three-membered cyclic phosphorus cations. Moreover, this suggests that the reaction is carried out using isolated species, which is not the case.

In addition, the confusion between the two phosphirenium and phosphiranium species is also apparent at several occasions in the manuscript and must be necessarily corrected.

Right from the abstract, the authors state that the opening is limited to intramolecular reactions leading only to the formation of cyclic phosphorus compounds. However, there are prior examples, mentioned later in the article (ref 50 and 54) but not clearly identified as such, of intermolecular reactions involving the opening of phosphiranium ions and phosphirenium ions by external nucleophiles. Ref 50 is cited to illustrate electrophilicity of phosphorus in threemembered phosphorus cationic species. And indeed, in this work, ring opening was achieved through addition of water and primary alcohols on phosphorus to produce the corresponding ring-opened products. This therefore clearly constitutes a precedent for the addition of a nucleophile to P, and the authors must therefore refer to it in an appropriate and unambiguous way. Concerning ref 54, a C-centered ring opening of isolated phosphiranium salts by anilines, leading to β -functionalized linear phosphines, is described. In connection with this work, the authors point out that the quoted reaction leads to the generation of phosphine oxides (line 54, page 3), whereas the work described by the French group produces unoxidized phosphines.

Two additional articles reporting electrophilic phosphination of alkynes promoted by Tf₂O should be added in ref 40:

(1) Hirano, K. and Miura, M. Development of New C-N and C-P Bond Formations with Alkenes and Alkynes Based on Electrophilic Amination and Phosphination. *J. Synth. Org. Chem. Jpn.* 2018, 1206-1214.

(2) Kazutoshi Nishimura, K.; Xu, S.; Nishii, Y. and Hirano, K. One-Step Synthesis of Benzophosphole Derivatives from Arylalkynes by Phosphenium-Dication-Mediated Sequential C-P/C-C Bond Forming Reaction. *Org. Lett.* 2023, 25, 1503-1508.

In this regard, ref 51 should not be disconnected from these latter major contributions since the chemistry developed therein is strongly correlated. This same reference should also be recalled when commenting on the results obtained from a competition reaction test using a propargylamine substrate (line 144, page 6).

Concerning mechanistic consideration, it would have been useful if the authors had mentioned whether they had actually observed the formation of phosphirenium intermediates, in particular by monitoring the reaction using ³¹P NMR, since phosphirenium ions exhibit a characteristic shift around -110 ppm. If not, control experiments should be carried out.

I also deplore the absence of discussion on the regioselectivity of the reaction observed with disymmetrical alkynes, especially since DFT calculations that constitute elements of discussion are succinctly given in the experimental section. At the very least, comments on this point should be included in the article and calculations should be further commented.

Besides, the literature is not always cited in an appropriate manner. I recommend that authors rigorously check the references of the all the articles cited. See below for representative examples of errors found:

-Ref 29 (line 249, page 9): No journal mentioned

-Ref 40 (line 274, page 10): The name of the Journal is wrong. The cited article is actually published in Chemistry a european Journal (*Chem. Eur. J.* 2018, 24, 13089 – 13092).

-In some references, the authors are not fully cited whereas they are in other cases.

A few misspelling or typing errors were also detected:

-Difunctionalization (« c » missing) in the title (line 2, page 1)

-« is occur » should be replaced by « is occurring » (line 60, page 3)

-« condition » should be replaced by « conditions » (line 97, page 4)

-« cynao » should be replaced by « cyano » and « pyrro » by « pyrrolo groups » (line 151, page 6)

-« iodines » should be replaced by « iodine » (line 181, page 8)

-« reagents » should be replaced by « reactivity » (line 186, page 8)

-« phosphorhole » should be replaced by « phosphole » (line 188, page 8)

In summary, I could not recommend the current manuscript for publication as it is. However, since the work presented in the article is interesting and should be of interest for synthetic chemists, I believe that it could be reconsidered for publication in Nature Communications provided that major revisions are made to the article

Reviewer #2 (Remarks to the Author):

Chi and coworkers present a method for the preparation of E-iodo-vinylphosphine oxides using elemental iodine as the source of I.

The manuscript is well laid out and easy to read.

The reaction operates over a range of phosphine oxide starting materials, with several sterically and electronically interesting and diverse phosphine oxides used. In this regard the authors have gone beyond a routine substrate scope. This is also demonstrated in the range of alkynes probed in the reaction.

The authors report isolated yields for their products, which is great, but this actually negatively impacts the impression of the research, which half of Figure 1 being yields that are 50% or less... 4 of the yields are in the low 30% range. It would be of value to readers for the authors to report a spectroscopic yield of these products and to comment on what else forms e.g. is the remaining 50-70% across these substrates (3f, h, i, j, k, m, n) unreacted starting material?

For the mechanistic studies- this reviewer feels that it is generally accepted that secondary phosphine oxide + Tf₂O + alkyne generates, as termed by the authors, QPrS and other research groups have isolated these 3-membered rings. So although their studies confirm this (Fig4a), this does not bring any particularly new insight.

Fig 4b- is there evidence for in situ QPrS with a chloride counterion? This study, in its current form, does not rule out another addition mechanism taking place for the chlorophosphine.

Fig4c- is this not a repeat of Miura's work (which we know proceed through the 3 membered QPrS) and therefore does not add anything new? Maybe I have misunderstood what the authors are trying to demonstrate here?

Page 6, line 156/157- some references to back up this statement are needed.

Figure 6 (and associated discussion)- one could argue that this method is less economical than the methods from Chen et al. and from Liu et al. because their routes simply involve the reaction of secondary phosphine oxide with alkyne, rather than the need of an iodide to be introduced and then lost- a more critical evaluation of their own work is needed here for context.

All in all, there's some interesting synthesis presented here, but there are details lacking and poor representation of the state-of-the-art in this field. Both aspects need worked on.

Point-by-point Response to the Reviewers' Comments

Dear reviewers:

We sincerely thank you for your invaluable comments and professional advice. Your insights have significantly enhanced the academic rigor of our article. In response to your suggestions and requests, we have carefully made the necessary corrections and modifications to the manuscript. We believe these changes have improved the quality of our work. Furthermore, we would like to provide the following details:

Responses to Reviewer #1:

- 1) The manuscript by Chi, et al describes the synthesis of β -iodo vinylphosphine oxides through activation of secondary phosphine oxides with Tf_2O , followed by addition on disubstituted alkynes and opening of the resulting phosphirenium ions intermediates with iodine. The article is generally well written and the supporting information is well done with all new compounds suitably characterized and described. The substrate scope has been studied thoroughly and the methodology is found to be widely applicable. Additional examples from non-aryl phosphine oxides (only one example shown) might nevertheless have been beneficial. Without any doubt, the manuscript is a consistent and interesting paper, however, I consider that major concerns should be addressed prior to publication.

Our Response: We are profoundly grateful to the reviewer for the essential criticisms and valuable suggestions on our manuscript. We also thank the reviewer for the affirmation of this work.

- 2) The title is not appropriate: It promotes the use of quaternary phosphiranium salts while the chemistry presented in the manuscript is not based on, nor involves, the formation of phosphiranium ions which are strictly speaking unsaturated three-membered cyclic phosphorus cations. Moreover, this suggests that the reaction is carried out using isolated species, which is not the case.

Our Response: We thank the reviewer for the careful reading and apologize for this impropriety. We have realized that this expression is not suitable, and we have changed [*Quaternary Phosphiranium Salts: Reshape Alkynes with P-Centered Ring Opening for Difunctionalization*] to [*Regio- and Stereoselective Access to Highly Substituted Vinylphosphine Oxides via Metal-Free Electrophilic Phosphonoiodination of Alkynes*].

- 3) In addition, the confusion between the two phosphirenium and phosphiranium species is also apparent at several occasions in the manuscript and must be necessarily corrected.

Our Response: We have clearly recognized the use of phosphirenium and phosphiranium, and corrected *phosphiranium* to *phosphirenium* throughout the revised manuscript.

- 4) Right from the abstract, the authors state that the opening is limited to intramolecular reactions leading only to the formation of cyclic phosphorus compounds. However, there are prior examples, mentioned later in the article (ref 50 and 54) but not clearly identified as such, of intermolecular reactions involving the opening of phosphiranium ions and phosphirenium ions by external nucleophiles. Ref 50 is cited to illustrate electrophilicity of phosphorus in threemembered phosphorus cationic species. And indeed, in this work, ring opening was achieved through addition of water and primary alcohols on phosphorus to produce the corresponding ring-opened products. This therefore clearly constitutes a precedent for the addition of a nucleophile to P, and the authors must therefore refer to it in an appropriate and unambiguous way.

Our Response: According to recent reports, there are two ring-opening pathways for phosphiranium ions and phosphirenium ions (or unsaturated three-membered cyclic phosphorus cations): the C-addition process and P-addition process. Among them, the C-addition ring-opening process is primarily limited to intramolecular reactions resulting in cyclic phosphorus compounds, and only a few intermolecular examples (ref 56, in the revised manuscript). In contrast, when external nucleophiles such as water and primary alcohols are used in the reaction, an intermolecular P-addition ring-opening process occurs, yielding hydrophosphorylated products.

Following the reviewer's suggestion, and in order to avoid the ambiguity that the ring-opening of phosphiranium ions and phosphirenium ions is limited to intramolecular reactions, we have explicitly described the intermolecular P-addition ring-opening process of phosphirenium triflates that occurs in the presence of external water in the revised manuscript (lines 53–54, page 3, ref 52). As follows:

“For example, Wild’s group reported that phosphirenium triflates can undergo ring-opening to obtain cis-hydrophosphorylated products in the presence of MeOH or H₂O.”

- 5) Concerning ref 54, a C-centered ring opening of isolated phosphiranium salts by anilines, leading to β -functionalized linear phosphines, is described. In connection with this work, the authors point out that the quoted reaction leads to the generation of phosphine oxides (line 54, page 3), whereas the work described by the French group produces unoxidized phosphines.

Our Response: We have changed [phosphine oxides] to [phosphine compounds] in the revised manuscript (line 55, page 3).

- 6) Two additional articles reporting electrophilic phosphination of alkynes promoted by Tf_2O should be added in ref 40:
 (1) Hirano, K. and Miura, M. Development of New C-N and C-P Bond Formations with Alkenes and Alkynes Based on Electrophilic Amination and Phosphination. *J. Synth. Org. Chem. Jpn.* 2018, 1206-1214.
 (2) Kazutoshi Nishimura, K.; Xu, S.; Nishii, Y. and Hirano, K. One-Step Synthesis of Benzophosphole Derivatives from Arylalkynes by Phosphenium-Dication-Mediated Sequential C-P/C-C Bond Forming Reaction. *Org. Lett.* 2023, 25, 1503-1508.

Our Response: We have added the related references about electrophilic phosphination of alkynes promoted by Tf_2O (ref 41 and 42). Please see page 10 of the revised manuscript, lines 298–302.

- 7) In this regard, ref 51 should not be disconnected from these latter major contributions since the chemistry developed therein is strongly correlated. This same reference should also be recalled when commenting on the results obtained from a competition reaction test using a propargylamine substrate (line 144, page 6).

Our Response: As mentioned by the reviewer, some of the results of the competition reaction using the propargylamine substrate are based on the work of Hirano and Miura, and the reference should be indicated here. We have added the reference in the revised manuscript (lines 142-144, page 5). As follows:

“Subsequently, when the substrate 5 was used in the reaction and iodine was added at the beginning, both the phosphoniodination product 6 and the cyclization product 7 were obtained simultaneously (Fig. 4b)⁵³.”

- 8) Concerning mechanistic consideration, it would have been useful if the authors had mentioned whether they had actually observed the formation of phosphirenium intermediates, in particular by monitoring the reaction using ^{31}P NMR, since phosphirenium ions exhibit a characteristic shift around -110 ppm. If not, control experiments should be carried out.

Our Response: In the revised manuscript, we have added new additions to the mechanistic experiments. By monitoring the reaction using ^{31}P NMR, we observed a peak of -104.4 ppm which is assigned to the characteristic shift of phosphirenium ions (lines 155-156, page 7). As follows:

*“To further confirm the existence of this intermediate in our reaction, we conducted an *in situ* ^{31}P NMR experiment before adding I_2 and observed a distinct peak of -104.4 ppm which is assigned to phosphirenium species A (Fig. 4c).”*

Figure 1 *In situ* ^{31}P NMR (Fig. 4c in the revised manuscript)

Procedure for *in situ* ^{31}P NMR of SPOs and alkynes: A 10 mL oven-dried sealed tube equipped with a magnetic stir bar was charged with secondary phosphine oxides **1a** (0.2 mmol, 1.0 equiv) and alkyne **2b** (0.4 mmol, 2.0 equiv). The tube was evacuated and backfilled with argon (three times), and then CDCl_3 (1.0 mL) was added sequentially via a syringe, followed by 2,6-di-*tert*-butylpyridine (0.4 mmol, 2.0 equiv), and Tf_2O (0.21 mmol, 1.05 equiv). The resulting mixture was stirred for 3 h at 60 °C. After cooling ambient temperature, the reaction solution was transferred to an NMR tube under argon and detected by ^{31}P NMR, and we observed a peak of -104.4 ppm which is assigned to the characteristic shift of phosphirenium ions.

- 9) I also deplore the absence of discussion on the regioselectivity of the reaction observed with disymmetrical alkynes, especially since DFT calculations that constitute elements of discussion are succinctly given in the experimental section. At the very least, comments on this point should be included in the article and calculations should be further commented.

Our Response: We are very grateful for the reviewer's suggestions. For the regioselectivity of disymmetrical alkynes, we have added the description about this part in the revised manuscript. Additionally, we explored the regioselectivity and *Z/E* selectivity of

this reaction in more depth by DFT calculations. Please see page 7 of the revised manuscript, lines 158–172. Detailed information (DFT calculation and experimental study) is provided in SI (page 8-15). Additions in the revised manuscript are as follows:

“Furthermore, density functional theory (DFT) calculations were used to explain the reaction selectivity. As depicted in Fig. 4d, the alkenyl carbon attached to the ethyl group in the phosphirenium triflate (**B**) demonstrates higher nucleophilicity, while in phosphirenium iodide (**C**), the nucleophilicity of this alkenyl carbon is reduced, meaning that **C** is more likely to be a precursor of ring-opening. Subsequently, we calculated the relative Gibbs free energies of intermediates **D**, **E** and **F** that may be produced by ring-opening of three-membered cyclic phosphorus cation intermediate to further explain the regioselectivity and Z/E-selectivity of this reaction (Fig. 4f). The relative free energy of intermediate **E** is 1.5 kcal/mol higher than that of intermediate **D**, suggesting a preferential addition of the iodine cation to the alkenyl carbon attached to the phenyl group. For Z/E-selectivity, the relative free energy of intermediate **F** is 4.6 kcal/mol higher than that of intermediate **D**. Therefore, **D** is thermodynamically more stable, which is consistent with the experimental observations. Moreover, despite our efforts, we were unable to locate the intermediate **G'** depicted in theory (see SI section 6.2.2). Only the three-membered ring intermediate **G** could be identified, suggesting that the ring-opening process may proceed in a concerted manner.”

Figure 2 Studies of the regioselectivity and Z/E selectivity (**Fig. 4d** and **4e** in the revised manuscript)

10) Besides, the literature is not always cited in an appropriate manner. I recommend that authors rigorously check the references of the all the articles cited. See below for representative examples of errors found:

-Ref 29 (line 249, page 9): No journal mentioned

Our Response: We have added journal of ref 29 in the revised manuscript (lines 271-273, page 10).

-Ref 40 (line 274, page 10): The name of the Journal is wrong. The cited article is actually published in Chemistry a european Journal (Chem. Eur. J. 2018, 24, 13089 – 13092).

Our Response: Revised.

-In some references, the authors are not fully cited whereas they are in other cases.

Our Response: In line with previous reference formats in Nature Communications, if an article has no more than five authors, we list all of them. However, if an article has more than five authors, we list only the first author, and the remaining authors are represented as “et al.”.

11) A few misspelling or typing errors were also detected:

-Difunctionalization (« c » missing) in the title (line 2, page 1)

-« is occur » should be replaced by « is occurring » (line 60, page 3)

-« condition » should be replaced by « conditions » (line 97, page 4)

-« cynao » should be replaced by « cyano » and « pyrro » by « pyrrolo groups » (line 151, page 6)

-« iodines » should be replaced by « iodine » (line 181, page 8)

-« reagents » should be replaced by « reactivity » (line 186, page 8)

-« phosphorhole » should be replaced by « phosphole » (line 188, page 8)

Our Response: Revised.

12) In summary, I could not recommend the current manuscript for publication as it is. However, since the work presented in the article is interesting and should be of interest for synthetic chemists, I believe that it could be reconsidered for publication in Nature Communications provided that major revisions are made to the article

Our Response: Thank you again for your valuable comments and suggestions, and we hope that this revision is suitable for publication.

Responses to Reviewer #2

- 1) Chi and coworkers present a method for the preparation of E-iodo-vinylphosphine oxides using elemental iodine as the source of I. The manuscript is well laid out and easy to read. The reaction operates over a range of phosphine oxide starting materials, with several sterically and electronically interesting and diverse phosphine oxides used. In this regard the authors have gone beyond a routine substrate scope. This is also demonstrated in the range of alkynes probed in the reaction.

Our Response: We thank the reviewer for the time and effort in reviewing this manuscript. We also thank the reviewer for recognizing this work. The following is our point-by-point response to the reviewer comments.

- 2) The authors report isolated yields for their products, which is great, but this actually negatively impacts the impression of the research, which half of Figure 1 being yields that are 50% or less... 4 of the yields are in the low 30% range. It would be of value to readers for the authors to report a spectroscopic yield of these products and to comment on what else forms e.g. is the remaining 50-70% across these substrates (3f, h, i, j, k, m, n) unreacted starting material?

Our Response: For some examples with isolated yields below 50%, we have supplemented the NMR yields, indicating both the NMR yields (in parentheses) and the isolated yields below:

Figure 3 Scope of secondary phosphine oxides (**Fig. 2** in the revised manuscript)

It is noteworthy that in the analysis of TLC and ^{31}P NMR of the reaction system before purification, we did not observe any remaining SPOs. We observed that when the benzene rings had substituents at the *ortho*-position, trivalent phosphorus (**3m'** and **3n'**) was predominantly obtained as the major product, with a certain amount of pentavalent phosphorus. This observation may be attributed to steric hindrance that leads to the C-centered ring-opening. And we didn't observe trivalent phosphorus products when using other SPOs as substrates. Thank you very much again for raising this point, which enabled us to correct this error before the article was officially published. In the revised manuscript, we have provided a detailed description of Fig. 2 (lines 107-115, page 4). As follows:

“The steric effect had a certain impact on the reaction, leading to decreased yields when the benzene rings had substituents at the meta-position (**3f**, **3h-3i**). Substituents at the ortho-position of the benzene rings resulted in the major products being trivalent phosphorus (**3m'** and **3n'**), with a certain amount of pentavalent phosphorus. This result may be attributed to steric hindrance that leads to the C-centered ring-opening. And we didn't observe trivalent phosphorus products when using other SPOs as substrates. To determine the Z/E configuration of trivalent phosphorus products, we chose **3m'** as an example. It was oxidized to pentavalent phosphorus by H₂O₂, and the configuration was subsequently confirmed through X-ray crystallographic analysis (CCDC no. 2300801).”

- 3) For the mechanistic studies- this reviewer feels that it is generally accepted that secondary phosphine oxide + Tf₂O + alkyne generates, as termed by the authors, QPrS and other research groups have isolated these 3-membered rings. So although their studies confirm this (Fig4a), this does not bring any particularly new insight.

Our Response: As pointed out by the reviewer, it is generally accepted that QPrS can be obtained from secondary phosphine oxides and alkynes in the presence of Tf₂O. However, most of the examples reported so far have primarily resulted in the formation of cyclic phosphorus compounds (through intramolecular C-addition ring-opening process) and hydrophosphorylated products (through intermolecular P-addition ring-opening process). In this work, we have successfully synthesized β -functionalized vinylphosphine oxides instead of hydrophosphorylated products during the intermolecular P-addition ring-opening process of QPrS by combining suitable nucleophilic and electrophilic reagents. **Considering the difference between this reaction and previous work, we conducted isotope labeling experiments to confirm that the phosphoniodination reaction also experiences three-membered cyclic phosphorus cation intermediate.** These experiments demonstrated that the oxygen in the product is derived from water during the quenching process. And we suggest that the strategy of combining electrophilic and nucleophilic reagents may open a new avenue for the synthesis of unique vinylphosphine oxides through QPrS.

- 4) Fig 4b- is there evidence for in situ QPrS with a chloride counterion? This study, in its current form, does not rule out another addition mechanism taking place for the chlorophosphine.

Our Response: Thank you for bringing up this important issue, which motivated us to explore this experiment further. When chlorophosphine and alkyne were used in the reaction in the absence of Tf₂O, the corresponding product **3b** was obtained in 23% yield (in the original manuscript, or in **Fig. 4a below**). To further verify whether the reaction proceeds through a QPrS intermediate when using chlorophosphine as a substrate, we conducted the reaction in the absence of I₂ (**Fig. 4b**). Unfortunately, we didn't observe the corresponding phosphorus peak of QPrS with a chloride counterion in the *in situ* ³¹P NMR. This suggests that the reaction involving chlorophosphine might not proceed through a three-membered cyclic phosphorus cation intermediate. As a result, we have removed this experiment from the revised manuscript.

Figure 4 Reaction with Ph₂P-Cl as substrate (a) and *in situ* ³¹P NMR of Ph₂P-Cl with alkynes (b)

Procedure for Ph₂P-Cl as substrate: A 10 mL oven-dried sealed tube equipped with a magnetic stir bar was charged with chlorodiphenyl phosphine (0.2 mmol, 1.0 equiv). The tube was evacuated and backfilled with argon (three times) and then CHCl₃ (1.0 mL) was added sequentially via a syringe, followed by 1-phenyl-1-propyne (0.4 mmol, 2.0 equiv) and 2,6-di-*tert*-butylpyridine (0.4 mmol, 2.0 equiv) were added by a syringe. The resulting mixture was stirred for 3 h at 60 °C, and then I₂ (0.4 mmol, 2.0 equiv) was added and stirred for an additional 18 h at 60 °C. After cooling to ambient temperature, sat. NaHCO₃ aq (5.0 mL) was added and the resulting mixture was extracted with DCM (3 x 10 mL). The organic layer was washed with sat. Na₂S₂O₃ aq and brine, dried over MgSO₄, and then the volatiles were removed under reduced pressure. The residue was purified by flash column chromatography on silica gel to give the desired product **3b** (23% yield).

Procedure for *in situ* ^{31}P NMR of $\text{Ph}_2\text{P}-\text{Cl}$ and alkynes: A 10 mL oven-dried sealed tube equipped with a magnetic stir bar was charged with chlorodiphenylphosphine (0.2 mmol, 1.0 equiv). The tube was evacuated and backfilled with argon (three times). Subsequently, CDCl_3 (1.0 mL) was added sequentially via a syringe, followed by the addition of 1-phenyl-1-propyne (0.4 mmol, 2.0 equiv) and 2,6-di-*tert*-butylpyridine (0.4 mmol, 2.0 equiv). The resulting mixture was stirred for 21 h at 60 °C. After cooling to ambient temperature, the reaction solution was transferred to an NMR tube under argon and detected by ^{31}P NMR, while no obvious characteristic peak of QPrS with a chloride counterion was found.

- 5) Fig 4c- is this not a repeat of Miura's work (which we know proceed through the 3 membered QPrS) and therefore does not add anything new? Maybe I have misunderstood what the authors are trying to demonstrate here?

Our Response: Miura's Group has reported that in the absence of iodine, substrate **5** can yield the cyclic phosphorus compounds **7** through a three-membered cyclic phosphorus cation intermediate (Fig. 5a, *J. Am. Chem. Soc.* **139**, 6106-6109, (2017)). Building on this foundation, we introduced iodine into the reaction system to confirm that the phosphonoiodination reaction might also involve a three-membered cyclic phosphorus cation intermediate. **The resulting observation of both products 6 and 7 supports the idea that they are generated through a competitive reaction between the P-addition and C-addition processes of this intermediate (Fig. 5b).** To make our expression clearer, we have re-described the experiment in the revised manuscript (lines 142-145, page 5) and (lines 154-155, page 7). As follows:

“Subsequently, when the substrate **5** was used in the reaction and iodine was added at the beginning, both the phosphonoiodination product **6** and the cyclization product **7** were obtained simultaneously (Fig. 4b)⁵³. This suggests that the phosphonoiodination reaction may also undergo a three-membered cyclic phosphorus cation intermediate, and that these products result from a competitive reaction between the P-addition and C-addition process of this intermediate.”

Figure 5 Comparison of Miura's work and our work (Fig. 4b in the revised manuscript)

- 6) Page 6, line 156/157- some references to back up this statement are needed.

Our Response: Thank you for your suggestion. For some of these products, alternative methods of synthesis can be found (*ACS Catal.* **2018**, *8*, 10599 - 10605; *Org. Lett.* **2018**, *20*, 3341 – 3344), and we have deleted this statement for a more rigorous presentation.

- 7) Figure 6 (and associated discussion)- one could argue that this method is less economical than the methods from Chen et al. and from Liu et al. because their routes simply involve the reaction of secondary phosphine oxide with alkyne, rather than the need of an iodide to be introduced and then lost- a more critical evaluation of their own work is needed here for context.

Our Response: Thank you for bringing this issue to our attention. Indeed, for the synthesis of benzo[*b*]phosphole oxides, our approach may not be the optimal choice. But our method offers an alternative route for the direct generation of alkenyl radicals containing vinylphosphine oxide groups under metal-free conditions. It's worth noting that these phosphonoiodination products can serve as valuable intermediates for constructing a wide range of phosphorus-containing compounds. And we have slightly revised description of Figure 6 in the revised manuscript (lines 191-201, pages 8). As follows:

“In addition to transition metal-catalyzed coupling reactions, C-I bonds can also be transformed by other types of reactions. According to the literature, homolytic cleavage of C-I bonds in the presence of strong electron-absorbing groups can generate

corresponding iodine radicals and carbon radicals.^{23, 57-61} Here, we achieved a homogeneous cleavage of the C-I bond in the presence of 8-hydroxyquinoline and Cs₂CO₃, and the resulting carbon radical can undergo radical cyclization to give the benzo[b]phosphole oxides. As shown in Fig. 6, benzo[b]phosphole oxides with a variety of alkyl, aryl and heteroaryl groups on the backbone can be obtained in excellent yields (**15a-15i**). When there was methoxyphenyl substitution at the α -position of the carbon radical, the yield of the target product decreased significantly (**15j**), meanwhile the by-product ketone **16** was obtained in 31% yield. Besides, we are also exploring other transformations for this alkenyl radical containing a vinylphosphine oxide group.”

- 8) All in all, there's some interesting synthesis presented here, but there are details lacking and poor representation of the state-of-the-art in this field. Both aspects need worked on.

Response: Thank you again for taking time to review our manuscript. We appreciate your feedback regarding the level of detail and the representation of the state-of-the-art in the field. We have taken your comments seriously and added more details as well as DFT calculations to explain the possible mechanism, regioselectivity and Z/E-selectivity of this reaction in the revised manuscript. Please see page 5 of the revised manuscript, lines 142–145, and page 7, lines 154–172. Detailed information (DFT calculation and experimental study) is provided in SI (page 8-15). And we hope that the revised manuscript will convince you that our work is suitable for publication.

Reviewers' Comments:

Reviewer #1:

Remarks to the Author:

On the whole, the authors have seriously taken into account the remarks made during the first evaluation and have made all the requested corrections and additional experiments, thus improving their manuscript.

In these circumstances, I can now recommend the article submitted by Chi and colleagues for publication in Nature Communications.

Reviewer #2:

Remarks to the Author:

The authors have addressed many of the original questions raised by the reviewers. However, some queries were not answered in a satisfactory way or the new information provided raises more questions.

1. Some spectroscopic yields have been provided, but since these are still not able to explain the course of the reaction i.e. is the remainder of the spectroscopic yield for the compounds starting material? Or side-product?
2. No spectroscopic yield is provided for half of Figure 2 and all of Figure 3. How can the authors report such low yields (e.g. 4j, 9%) and not comment on the make-up of the reaction mixture?
3. substrates like 4j, 4k do help to provide evidence for their mechanism, but only if we know the make-up of the rest of the reaction mixture.
4. The authors have tried to use calculations to prove their mechanism. However, the formation of the phosphirenium iodide would require TfOI to form- is this sensible? This seems highly unlikely. Why can't the triflate phosphirenium not just react via the P-centred mechanism with I²⁻- why is a change in counter ion needed?
5. To provide adequate support for Fig 4d and e, single point energy calculations are not enough and transition state calculations are needed.
6. Based on basic undergraduate chemistry, we would assume that species D and E are more stable than F (Fig. 4e), but calculating this without transition states is not acceptable.
7. Work from Grimme (PCCP 2017) has shown that double hybrid functionals perform better than standard/popular functionals for main group calculations- have the authors checked they are using the best method, as opposed to the most 'popular' one (as stated by the authors).

That the synthesis is interesting, there is no doubt. However, to be suitable for Nature Commun. a more exacting analysis of their synthesis results is needed, in the form of spectroscopic yield and accounting for non-product yields. New additions to the paper, for this reviewer, detract from the good quality synthetic work carried out here.

REVIEWER COMMENTS

Reviewer #1 (Remarks to the Author):

On the whole, the authors have seriously taken into account the remarks made during the first evaluation and have made all the requested corrections and additional experiments, thus improving their manuscript.

In these circumstances, I can now recommend the article submitted by Chi and colleagues for publication in Nature Communications.

Reviewer #2 (Remarks to the Author):

The authors have addressed many of the original questions raised by the reviewers. However, some queries were not answered in a satisfactory way or the new information provided raises more questions.

1. Some spectroscopic yields have been provided, but since these are still not able to explain the course of the reaction i.e. is the remainder of the spectroscopic yield for the compounds starting material? Or side-product?
2. No spectroscopic yield is provided for half of Figure 2 and all of Figure 3. How can the authors report such low yields (e.g. 4j, 9%) and not comment on the make-up of the reaction mixture?
3. substrates like 4j, 4k do help to provide evidence for their mechanism, but only if we know the make-up of the rest of the reaction mixture.
4. The authors have tried to use calculations to prove their mechanism. However, the formation of the phosphirenium iodide would require TfOI to form- is this sensible? This seems highly unlikely. Why can't the triflate phosphirenium not just react via the P-centred mechanism with I²⁻- why is a change in counter ion needed?
5. To provide adequate support for Fig 4d and e, single point energy calculations are not enough and transition state calculations are needed.
6. Based on basic undergraduate chemistry, we would assume that species D and E are more stable than F (Fig. 4e), but calculating this without transition states is not acceptable.
7. Work from Grimme (PCCP 2017) has shown that double hybrid functionals perform better than standard/popular functionals for main group calculations- have the authors checked they are using the best method, as opposed to the most 'popular' one (as stated by the authors).

That the synthesis is interesting, there is no doubt. However, to be suitable for Nature Commun. a more exacting analysis of their synthesis results is needed, in the form of spectroscopic yield and accounting for non-product yields. New additions to the paper, for this reviewer, detract from the good quality synthetic work carried out here.

Point-by-point Response to the Reviewers' Comments

Dear reviewers:

We sincerely thank you again for taking time to review our manuscript. Your insights have significantly enhanced the academic rigor of our article. According to your suggestions, we have carefully provided the necessary additions to the manuscript, and we would like to provide the following details:

Responses to Reviewer #1

On the whole, the authors have seriously taken into account the remarks made during the first evaluation and have made all the requested corrections and additional experiments, thus improving their manuscript.

In these circumstances, I can now recommend the article submitted by Chi and colleagues for publication in Nature Communications.

Our Response: We appreciate your time and advice in reviewing our work and are very pleased that our work has been recognized by you!

Responses to Reviewer #2

- 1) The authors have addressed many of the original questions raised by the reviewers. However, some queries were not answered in a satisfactory way or the new information provided raises more questions. Some spectroscopic yields have been provided, but since these are still not able to explain the course of the reaction i.e. is the remainder of the spectroscopic yield for the compounds starting material? Or side-product?

Our Response: Thank you for your suggestion. We analyzed the crude reaction mixture by NMR. These results are as follows: For alkynes, we observed the formation of diiodination product and the remaining of starting alkynes and we analyzed the transformations and residuals of alkynes with examples of **3a**, **3h**, **4j**, **4k**, **4l** and **4v**. As seen in **Table 1**, the sum of conversions and residuals of alkynes after the reaction essentially matches the initially added amount of alkynes.

Table 1: Alkynes' conversions and residuals

Entry	Product	Diiodination	Starting Alkynes	Total alkynes converted and remaining
1	3a , 0.164 mmol	0.13 mmol	0.1 mmol	0.394 mmol, 99%
2	3h , 0.1 mmol	0.25 mmol	0.04 mmol	0.39 mmol, 98%
3	4j , 0.024 mmol	0.24 mmol	0.13 mmol	0.394 mmol, 99%
4	4k , 0.028 mmol	0.26 mmol	0.11 mmol	0.398 mmol, > 99%
5	4l , 0.09 mmol	0.2 mmol	0.11 mmol	0.4 mmol, > 99%
6	4v , 0.042 mmol	0.35 mmol	/	0.392 mmol, 98%

For SPOs, we only observed the formation of phosphoric anhydrides (~28.5 ppm) and products in the ³¹P NMR spectrum of the crude mixture, and the phosphoric anhydrides were also detected by HRMS. **No starting SPOs were detected.** In **Figure 1** (shown below), we provided the crude ³¹P NMR spectra with examples of **3a**, **4k** and **3h**, with Ph₃PO added as a reference. Additionally, the hydrophosphorylated product (HRMS (ESI) calcd for C₂₄H₂₆OP⁺ [M+H]⁺ 361.1716, found 361.1716) and di-*m*-tolylphosphinic acid (HRMS (ESI) calcd for C₁₄H₁₆O₂P⁺ [M+H]⁺ 247.0882, found 247.0883) were also detected by HRMS in the crude mixture of **3h**. However, their amounts are so small that it is difficult to observe them in the ³¹P NMR spectrum.

Figure 1: Crude ^{31}P NMR spectra of reaction

Procedure for crude ^{31}P NMR spectra of reaction: A 10 mL oven-dried sealed tube equipped with a magnetic stir bar was charged with SPOs (0.2 mmol, 1.0 equiv). The tube was evacuated and backfilled with argon (three times) and then CHCl_3 (1.0 mL) was added sequentially via a syringe, followed by Tf_2O (0.21 mmol, 1.05 equiv), 2,6-di-*tert*-butylpyridine (0.4 mmol, 2.0 equiv) and alkynes (0.4 mmol, 2.0 equiv) were added by a syringe. The resulting mixture was stirred for 3 h at 60 °C, and then I_2 (0.4 mmol, 2.0 equiv) was added and stirred for an additional 18 h at 60 °C. After cooling to ambient temperature, sat. NaHCO_3 aq (5.0 mL) was added and the resulting mixture was extracted with DCM (3 x 10 mL). The organic layer was washed with sat. $\text{Na}_2\text{S}_2\text{O}_3$ aq and brine, dried over MgSO_4 , and then the volatiles were removed under reduced pressure. Ph_3PO was added to the residue as a reference, dissolved with CDCl_3 , and then the solution was transferred to an NMR tube and analyzed by ^{31}P NMR.

- 2) No spectroscopic yield is provided for half of Figure 2 and all of Figure 3. How can the authors report such low yields (e.g. 4j, 9%) and not comment on the make-up of the reaction mixture?

Our Response: According to the previous suggestions from the reviewer, we provided the spectroscopic yields for examples with yields below 50% in both Figure 2 and Figure 3. In the analysis of ^{31}P NMR spectra for the crude mixture of 4j and 4k, only phosphoric anhydride (~28.5 ppm) and products were observed. As for the composition of the reaction mixture, please refer to the response provided in Q1. Additionally, we have added the analysis of the make-up of the reaction mixture in the revised SI (pages 9-11).

- 3) substrates like 4j, 4k do help to provide evidence for their mechanism, but only if we know the make-up of the rest of the reaction mixture.

Our Response: From the ^{31}P NMR spectrum of the crude mixture, we only observed the formation of phosphoric anhydrides and products. For **4j** and **4k**, the presence of CF_3 makes it less likely to form the three-membered cyclic phosphorus cation intermediate, and we only observed the formation of phosphoric anhydride when alkynes were removed, supporting our proposed mechanism to some extent. Additionally, we have added the explanations for the low yields of **4j** and **4k** in the revised manuscript (lines 124–127, page 5).

- 4) The authors have tried to use calculations to prove their mechanism. However, the formation of the phosphirenium iodide would require TfOI to form- is this sensible? This seems highly unlikely. Why can't the triflate phosphirenium not just react via the P-centred mechanism with I $^-$ - why is a change in counter ion needed?

Our Response: Considering the reviewer's concerns, we conducted DFT calculations based on both phosphirenium iodide and phosphirenium triflate in the revised manuscript (lines 158–169, page 6) and SI (pages 11-43). The results indicated that calculations based on phosphirenium iodide support the generation of the product both kinetically and thermodynamically, **while the calculations on the latter revealing that the energy barrier for the transition states is higher than that of former (about twice). Therefore, we think that the presence of the iodine anions increases the driving force for the ring-opening.** Additionally, we think that TfO^- and I^+ are exist separately as free forms in the reaction system, and the 2,6- di-*tert*-butylpyridine present in the system may have the ability to capture iodine cations (J. Am. Chem. Soc. **2016**, *138*, 9853-9863.; Chem. Soc. Rev. **2020**, *49*, 2688-2700.; ChemSusChem, **2021**, *14*, 738-744.).

- 5) To provide adequate support for Fig 4d and e, single point energy calculations are not enough and transition state calculations are needed.

Our Response: Thank you for your reminding. We have supplemented transition state calculations in the revised manuscript (lines 158–169, page 6) and SI (pages 11-43). As follows:

Figure 2. DFT calculations based on phosphirenium iodide (Fig. 4d in the revised manuscript)

- 6) Based on basic undergraduate chemistry, we would assume that species D and E are more stable than F (Fig. 4e), but calculating this without transition states is not acceptable.

Our Response: According to the reviewer's suggestions, we have supplemented new calculations in the revised manuscript and SI (please see our answer to Q5 or pages 11-43 in the supporting information).

- 7) Work from Grimme (PCCP 2017) has shown that double hybrid functionals perform better than standard/popular functionals for main group calculations- have the authors checked they are using the best method, as opposed to the most 'popular' one (as stated by the authors).

Our Response: To address the reviewer's concern and ensure the reliability of our chosen method, various methods, including M06-2X, B3LYP-D3 and DSD-PBEP86 were additionally employed for recomputing the single-point energies of the key transition state **A-TS1** and intermediate **M0**. The computed results have been compiled and presented in **Table 2**. There are small differences in the energies calculated by different methods, and the computed results by M06-2X can reasonably explain the experimental observation, indicating that the selected DFT method should be suitable for this kind of system.

Table 2. Comparison of ΔG for A-TS1 and M0 with different methods (unit: kcal/mol)

Method	M06-2X	B3LYP-D3	DSD-PBEP86
ΔG^\ddagger (A-TS1 – M0)	15.7	13.5	15.9

- 8) That the synthesis is interesting, there is no doubt. However, to be suitable for Nature Commun. a more exacting analysis of their synthesis results is needed, in the form of spectroscopic yield and accounting for non-product yields. New additions to the paper, for this reviewer, detract from the good quality synthetic work carried out here.

Response: Thank you again for your thoughtful review of our manuscript. We have carefully considered your comments and made every effort to address them, and we hope that the new additions and changes in the revised manuscript will convince you that our work is suitable for publication.

Reviewers' Comments:

Reviewer #2:

Remarks to the Author:

The authors have addressed all the reviewers comments.

It is difficult, as a non-computational chemist, to assess whether the steps to test basis sets and possible transition states are sufficient. Ideally, if computational work is involved in a manuscript, the editors should be seeking computational reviewers.

However, in my capacity as a synthetic chemist, the comments have been addressed and the manuscript is suitable for publication in Nature Commun.

Reviewer #3:

Remarks to the Author:

In order to justify the choice of level of theory for the DFT calculations, the authors compute the Gibbs energy difference between species M0 and A-TS1 obtaining values in the 13.5 – 15.9 kcal/mol range. Given that DFT calculations are used to justify the chemoselectivity of the reported reactivity, Table S8 should be improved by adding the values for the energy difference between M0 and B-TS1, C-TS1, D-TS1, respectively, to ensure that the chemoselectivity is kept unaltered despite the level of theory employed.

From the computational details provided, it is hard to replicate those calculations. It is not clear whether pseudopotentials were used for iodine atoms. Moreover, authors should clarify if solvent effects were considered in the optimization process or in later single point calculations. Authors should also include more detailed information on the characterization of stationary points (i.e. frequency calculations).

As minor points, language should be double check. I would like to highlight a spelling error in the title (vinyphosphine should be replaced by vinylphosphine), and the repeated use of iodine anion instead of iodide.

The references section is an example again that the authors did not put all efforts on this document, with full journal titles mixed with abbreviations, strange commas. And in the author contributions for example again simply the "DFT calculation" those guys did? Only one? The SI text could be improved, take for instance with repetitive use of words, errors such as "has the much lower energy than the other three intermediates." The energy profiles use relative energies, this is right, but just look at Figure S4, the differences between the four different TS2 is completely out of scale, this is on purpose to show energy differences are in the same range? In Figure S5 again but it is more schematic (2 lines between INT and M2 would help).

Even though we must consider that still the manuscript has "flaws", they correspond to minor points, actually to format particularly. Thus, the paper after having addressed the previous concerns and those current errors that can be solved in simply 1-2 hours, the paper deserves publication in Nature Communications.

Reviewer #4:

Remarks to the Author:

REVIEWER COMMENTS

Reviewer #2 (Remarks to the Author):

The authors have addressed all the reviewers comments.

It is difficult, as a non-computational chemist, to assess whether the steps to test basis sets and possible transition states are sufficient. Ideally, if computational work is involved in a manuscript, the editors should be seeking computational reviewers.

However, in my capacity as a synthetic chemist, the comments have been addressed and the manuscript is suitable for publication in *Nature Commun.*

Reviewer #3 (Remarks to the Author):

In order to justify the choice of level of theory for the DFT calculations, the authors compute the Gibbs energy difference between species M0 and A-TS1 obtaining values in the 13.5 – 15.9 kcal/mol range. Given that DFT calculations are used to justify the chemoselectivity of the reported reactivity, Table S8 should be improved by adding the values for the energy difference between M0 and B-TS1, C-TS1, D-TS1, respectively, to ensure that the chemoselectivity is kept unaltered despite the level of theory employed.

From the computational details provided, it is hard to replicate those calculations. It is not clear whether pseudopotentials were used for iodine atoms. Moreover, authors should clarify if solvent effects were considered in the optimization process or in later single point calculations. Authors should also include more detailed information on the characterization of stationary points (i.e. frequency calculations).

As minor points, language should be double check. I would like to highlight a spelling error in the title (vinyphosphine should be replaced by vinylphosphine), and the repeated use of iodine anion instead of iodide.

The references section is an example again that the authors did not put all efforts on this document, with full journal titles mixed with abbreviations, strange commas. And in the author contributions for example again simply the “DFT calculation” those guys did? Only one? The SI text could be improved, take for instance with repetitive use of words, errors such as “has the much lower energy than the other three intermediates.” The energy profiles use relative energies, this is right, but just look at Figure S4, the differences between the four different TS2 is completely out of scale, this is on purpose to show energy differences are in the same range? In Figure S5 again but it is more schematic (2 lines between INT and M2 would help).

Even though we must consider that still the manuscript has “flaws”, they correspond to minor points, actually to format particularly. Thus, the paper after having addressed the previous concerns and those current errors that can be solved in simply 1-2 hours, the paper deserves publication in Nature Communications.

Point-by-point Response to the Reviewers' Comments

Dear reviewers:

We sincerely thank you again for taking time to review our manuscript. Your insights have significantly enhanced the academic rigor of our article. According to your suggestions, we have carefully provided the necessary additions to the details of the manuscript, and we would like to provide the following details:

Reviewer #2 (Remarks to the Author):

The authors have addressed all the reviewers comments.

It is difficult, as a non-computational chemist, to assess whether the steps to test basis sets and possible transition states are sufficient. Ideally, if computational work is involved in a manuscript, the editors should be seeking computational reviewers.

However, in my capacity as a synthetic chemist, the comments have been addressed and the manuscript is suitable for publication in *Nature Commun.*

Our Response: We appreciate your time and advice in reviewing our work, and we are very pleased that our work has been recognized by you!

Responses to Reviewer #3

- 1) In order to justify the choice of level of theory for the DFT calculations, the authors compute the Gibbs energy difference between species M0 and A-TS1 obtaining values in the 13.5 – 15.9 kcal/mol range. Given that DFT calculations are used to justify the chemoselectivity of the reported reactivity, Table S8 should be improved by adding the values for the energy difference between M0 and B-TS1, C-TS1, D-TS1, respectively, to ensure that the chemoselectivity is kept unaltered despite the level of theory employed.

Our Response: Thank you for your advice. We appreciate your attention to the justification of the level of theory used in our DFT calculations. We have conducted structural optimizations by using different methods, including M06-2X, B3LYP-D3 and DSD-PBEP86 to justify the chemoselectivity of the reported reactivity. The Gibbs energies of the transition states **B-TS1**, **C-TS1**, and **D-TS1** were compiled and presented in **Table 1 (Table S8, page 15 in Supplementary Information)**. **The computational results demonstrate that the pathway associated with transition state A-TS1 is consistently identified as the most energetically favorable pathway regardless of the computational methods employed.**

Table 1. Comparison of ΔG for **A-TS1**, **B-TS1**, **C-TS1**, and **D-TS1** optimized with different methods (unit: kcal/mol)

Method	M06-2X	B3LYP-D3	DSD-PBEP86
$\Delta G_{(\text{A-TS1})}$	0.0	0.0	0.0
$\Delta G_{(\text{B-TS1})}$	5.5	2.1	3.7
$\Delta G_{(\text{C-TS1})}$	1.4	2.0	3.0
$\Delta G_{(\text{D-TS1})}$	2.1	3.9	5.2

- 2) From the computational details provided, it is hard to replicate those calculations. It is not clear whether pseudopotentials were used for iodine atoms. Moreover, authors should clarify if solvent effects were considered in the optimization process or in later single point calculations. Authors should also include more detailed information on the characterization of stationary points (i.e. frequency calculations).

Our Response: Thank you for your insightful comments and suggestions. We have provided detailed computational details for different DFT methods in the **Source Data**. Correspondingly, we have provided a detailed description “Gaussian09 program⁹ was employed for the theoretical calculations. The geometry optimization was carried out by using the M06-2X method¹⁰. The 6-31G (d, p) basis set¹¹ was used for C, H, O, and P atoms, whereas the LANL2DZ pseudopotential basis set¹² was used for the I atom.

Solvent effects were considered using the integral equation formalism polarizable continuum model (IEF-PCM)¹³ with chloroform as the solvent.” in the supplementary information (page 12).

Additionally, we have provided detailed characterization of stationary points, including frequency calculations in the *Source Data*, to ensure the reliability of our study.

- 3) As minor points, language should be double check. I would like to highlight a spelling error in the title (vinyphosphine should be replaced by vinylphosphine), and the repeated use of iodine anion instead of iodide. The references section is an example again that the authors did not put all efforts on this document, with full journal titles mixed with abbreviations, strange commas. And in the author contributions for example again simply the “DFT calculation” those guys did? Only one? The SI text could be improved, take for instance with repetitive use of words, errors such as “has the much lower energy than the other three intermediates.”

Our Response: We are very grateful to the reviewers for their careful review. We've corrected the spelling mistake in the title. The term "iodine anion" has been replaced by "iodide" in the appropriate place. We have also double-checked the references section and changed "DFT calculation" to "DFT studies" in the author contributions. We have also changed the details suggested by the reviewers in the supplementary information, and all changes in the revised manuscript and supplementary information (pages 12-15) are highlighted in blue.

- 4) The energy profiles use relative energies, this is right, but just look at Figure S4, the differences between the four different TS2 is completely out of scale, this is on purpose to show energy differences are in the same range? In Figure S5 again but it is more schematic (2 lines between INT and M2 would help).

Our Response: Thank you for your reminding. To improve the aesthetics and rationalization of energy profiles, we have adjusted the ratio between the four different **TS2** and **M2** in **Figure S4** and *embellished Figure S3*, and we have also added 2 lines between **INT** and **M2** in **Figure S5** as suggested by the reviewer in the supplementary information (pages 13-14).

- 5) Even though we must consider that still the manuscript has “flaws”, they correspond to minor points, actually to format particularly. Thus, the paper after having addressed the previous concerns and those current errors that can be solved in simply 1-2 hours, the paper deserves publication in Nature Communications.

Response: Thank you again for your thoughtful review of our manuscript. We have carefully considered your comments and made every effort to address them, and we hope that the revised manuscript will convince you that our work is suitable for publication in *Nature Communications*.